# Eliminate Before Align: A Remote Sensing Image-Text Retrieval Framework with Keyword Explicit Reasoning

## ABSTRACT

Mountains of researches center around the Remote Sensing Image-Text Retrieval (RSITR), aiming at retrieving the corresponding targets based on the given query. Among them, the transfer of Foundation Models (FMs), such as CLIP, to remote sensing domain shows promising results. However, existing FM-based approaches neglect the negative impact of weakly correlated sample pairs and the key distinctions among remote sensing texts, leading to biased and superficial exploration of sample pairs. To address these challenges, we propose a novel Eliminate Before Align strategy with Keyword Explicit Reasoning framework (EBAKER) for RSITR. Specifically, we devise an innovative Eliminate Before Align (EBA) strategy to filter out the weakly correlated sample pairs to mitigate their deviations from optimal embedding space during alignment. Moreover, we introduce a Keyword Explicit Reasoning (KER) module to facilitate the positive role of subtle key concept differences. Without bells and whistles, our method achieves a one-step transformation from FM to RSITR task, obviating the necessity for extra pretraining on remote sensing data. Extensive experiments on three popular benchmark datasets validate that our proposed EBAKER method outperform the state-of-the-art methods with fewer training data. Our source code will be released soon.

## CCS CONCEPTS

• **Information systems** → **Information retrieval**; **Specialized information retrieval**; **Multimedia and multimodal retrieval**.

## KEYWORDS

Remote Sensing; Image-Text Retrieval; Foundation Model; Keyword Explicit Reasoning

## 1 INTRODUCTION

With the advancement of aerospace technology, remote sensing imagery has become increasingly accessible and finds wide applications in disaster monitoring [17], navigation [36], and agricultural production [33]. Among these applications, Remote Sensing Image-Text Retrieval (RSITR) stands as a foundational technique in remote sensing vision language domain [19], aiming to retrieve semantically similar images based on given text queries, and vice versa.

Recent research efforts have shifted towards RSITR, with the design of several effective methods leveraging from convolutional

Permission to make digital or hard copies of all or part of this work for personal or classroom use is granted without fee provided that copies are not made or distributed for profit or commercial advantage and that copies bear this notice and the full citation on the first page. Copyrights for components of this work owned by others than ACM must be honored. Abstracting with credit is permitted. To copy otherwise, or republish, to post on servers or to redistribute to lists, requires prior specific permission and/or a fee. Request permissions from permissions@acm.org.
*Conference'17, July 2017, Washington, DC, USA*
© 2024 Association for Computing Machinery.
ACM ISBN 978-x-xxxx-xxxx-x/YY/MM...$15.00
https://doi.org/10.1145/nnnnnnn.nnnnnnn

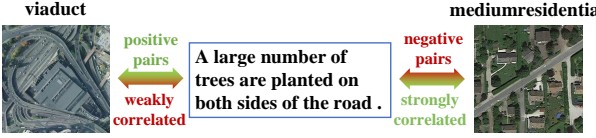

**Figure 1: Illustration of weakly correlated pairs. An image labeled as "A large number of trees are planted on both sides of the road" for a "viaduct" is weakly correlated, whereas "a medium residential" is strongly correlated but considered a negative image-text pair.**

neural networks [40–42] to Foundation Model (FM) [21, 45]. In the realm of RSITR, initial research efforts primarily revolve around CNN-based approaches [24, 25]. Abdullah et al. [1] pioneered the exploration of the RSITR problem by employing an average fusion strategy to attain robust representations. After that, plenty of CNN-based approaches [20, 27, 34, 38, 41, 42, 42] focused on refining alignment tailored to the characteristics of RSITR task. In recent years, with the flourishing development of FMs [4, 7, 15, 16, 29, 31] and their outstanding performance in various downstream tasks of image-text retrieval [10, 43, 44], such as text-based person search [35] and product search [3], researches in RSITR have pivoted towards the transfer from FM to Remote Sensing Image-Text Retrieval Model (RSITRM) [14, 45]. Yuan et al. [37] explored multiple Parameter-Efficient Fine-Tuning strategies to transfer CLIP to the remote sensing domain. Liu et al. [21] annotated and amalgamated multiple remote sensing datasets and compared the performance of different large-scale models such as CLIP [29], BLIP [16], and AL-BEF [18] in the remote sensing domain. Zhang et al. [45] proposed a 5M remote sensing dataset and achieved excellent performance by employing a two-step approach involving RS pretraining and downstream task fine-tuning to adapt CLIP to the remote sensing domain.

Regardless of whether they are traditional or FM-based approaches, they both necessitate more refined datasets. Finely and accurately annotated RSITR data will contribute to improving the performance of the model [45]. However, despite careful annotation, meaningless labeled image-text pairs still exist in the dataset [30]. Meaningless or weakly correlated positive image-text pairs may mislead the alignment of semantically relevant instances. For instance, as shown in Figure 1, an image of a "viaduct" may be labeled as "a large number of trees are planted on both sides of the road", which may be more relevant to a "mediumresidential". Such description, however, offers no benefit to the model. Therefore, it is worth exploring how to enable the model to autonomously eliminate the negative impact of such noise before fine-grained alignment.

In addition, existing FMs boost the development of RSITR on data volume [21, 45], which fail to grasp the core issue. The key of RSITR task lies not merely in increasing the quantity of positive and

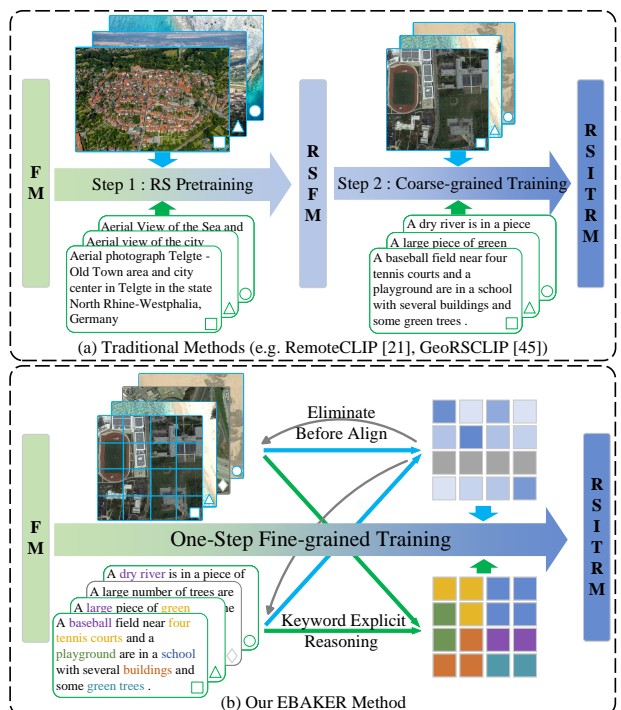

Figure 2: Comparison between our EBAKER and traditional methods on RSITR task. Traditional methods typically involve a two-step training strategy. (a) A substantial quantity of model-annotated remote sensing image-text pairs are utilized to transfer the Foundation Model (FM) into a Remote Sensing Foundation Model (RSFM), followed by the conversion of RSFM into a Remote Sensing Image-Text Retrieval Model (RSITRM) through additional coarse-grained contrastive learning on RSITR dataset. (b) We achieve a one-step transformation from FM to RSITRM, by implementing the Eliminate Before Align (EBA) strategy and the Keyword Explicit Reasoning (KER) module.

negative sample pairs in contrastive learning, but delving deeply into the key differences of sample pairs. Existing traditional and CLIP-based methods in RSITR predominantly utilize global features from vision and text encoders as their output, neglecting the key differences guided by the fine-grained features within remote sensing images.

The aforementioned two challenges have constrained the potential of FM, despite their significant advancements in various traditional multi-modal downstream tasks, as exemplified by CLIP [29] and BLIP [16]. The adaptation of FMs to RSITRMs still necessitates a substantial volume of remote sensing data, as illustrated in Figure 2. For instance, GeoRSCLIP [45] employs an additional 5M remote sensing image-text pairs and a two-step training to transfer CLIP to the RSITRM. This undoubtedly imposes an additional burden on training and presents challenging to achieving cost-effective performance growth.

To this end, we propose a novel framework called Eliminate Before Align strategy with Keyword Explicit Reasoning (EBAKER) for

achieving a one-step transition from FM to RSITRM, as illustrated in Figure 2 (b). Specifically, we introduce an innovative Eliminate Before Align (EBA) strategy to counteract the adverse effects of weakly correlated pairs. Additionally, a Keyword Explicit Reasoning (KER) module is introduced to facilitate the positive role of subtle key concept differences. We validate the efficacy of our proposed EBAKER method on three popular benchmark datasets, i.e., RSICD [23], RSITMD [38], and NWPU [5]. Extensive experiments demonstrate that EBAKER consistently outperforms state-of-the-art approaches.

Our contributions are summarized as follows:

- To achieve a one-step training from FM to RSITRM, we propose a novel Eliminate Before Align strategy with Keyword Explicit Reasoning framework (EBAKER), which aims at conducting fine-grained alignment through in-depth analysis of subtle distinctions and noise filtration. Our approach diverges from current state-of-the-art method GeoRSCLIP [45] by relying on a mere 4% of the training data and addressing the nuanced characteristics.
- To mitigate the negative impact of the weakly correlated pairs, we devise an innovative Eliminate Before Align (EBA) strategy. It enables autonomously eliminate the positive sample pairs with low global similarity before alignment, which promotes the accuracy of fine-grained contrastive learning and boosts the intrinsic confidence of the model.
- We introduce a Keyword Explicit Reasoning (KER) module, encouraging the model to predict subtle distinctions in key concepts within local features of remote sensing image-text pairs. This module facilitates fine-grained contrastive learning, enhancing the differentiation between extremely similar sample pairs.

## 2 METHOD

In this section, we present our proposed EBAKER framework, as illustrated in Figure 3. We begin with vision encoder, text encoder, and our keyword statistics and mask generation in Section 2.1. Next, we delve into our Eliminate Befora Align (EBA) strategy and Keyword Explicit Reasoning (KER) module in Section 2.2 and Section 2.3, respectively. Finally, the overall loss function is presented in Section 2.4.

### 2.1 Feature Extractor

*2.1.1 Vision Encoder.* Give the input image $I \in R^{(H \times W \times C)}$, we initially transform $I$ into $N = H \times W/P^2$ non-overlapping blocks of fixed size, where $N$ is the number of patches, $H$, $W$ ,and $C$ represent the height, width, and channel of the image respectively, $P$ represents the block size. Subsequently, all blocks are mapped to 1D tokens through a trainable linear projection. After incorpating positional encoding and an additional $[cls]$ token, the input block sequence is processed through $L$ layers of transformer blocks to establish the relationship between the input. Finally, all the features undergo linear projection, where $f_v^{cls}$ is transformed into the visual global feature $f_v^g$, and $\{f_v^1 \dots f_v^N\}$ represent the visual local features. The aformentioned process can be simplified as:

$$f_v^g, f_v^1 ... f_v^N = \varphi(I), \tag{1}$$

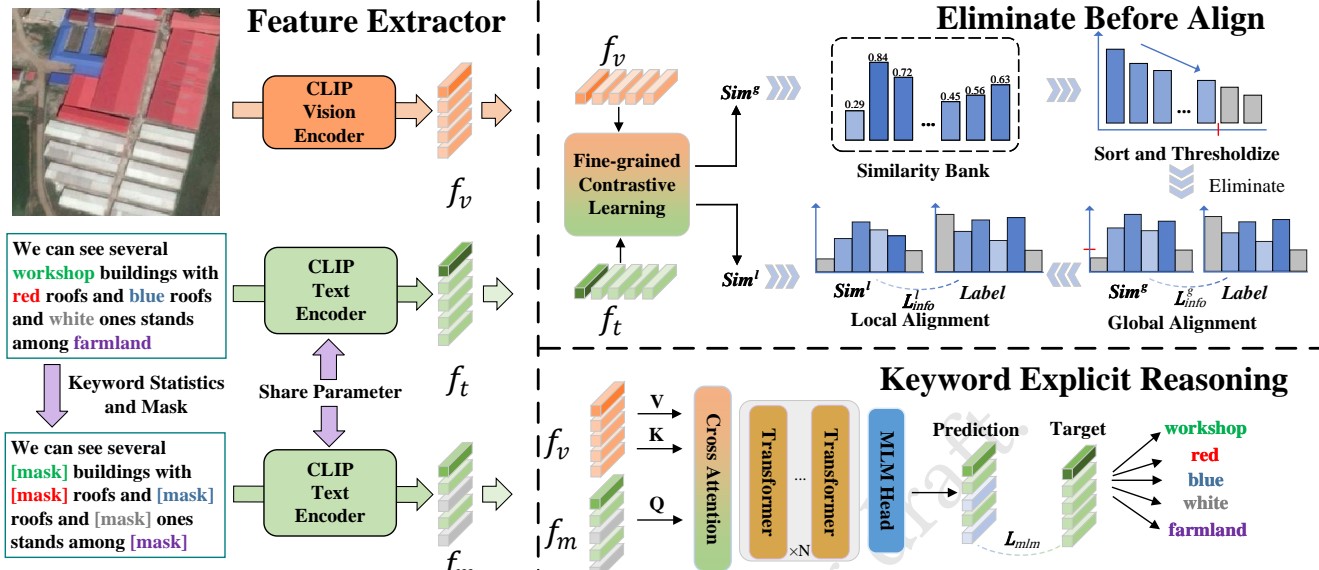

**Figure 3: An overview of our EBAKER approach, which consists of three parts. A. Feature Extractor: We employ CLIP as the encoders of both modalities, and conduct word frequency analysis for masking keywords. In the end, we obtain visual features, textual features, and masked textual features. B. Eliminate Before Align: Before alignment, we eliminate positive sample pairs with low global similarity to mitigate the negative impact of the weakly correlated pairs. C. Keyword Explicit Reasoning: We adopt a keyword prediction approach to facilitate the differentiation of subtle distinctions between remote sensing images.**

where $\varphi$ represents vision encoder of CLIP.

*2.1.2 Text Encoder.* For a given input text $T$, we utilize CLIP text encoder to extract representations. Initially, we tokenize the input text by lower-cased Byte Pair Encoding (BPE) with a vocabulary size of 49,152. The text description is surrounded by $[SOS]$ and $[EOS]$ tokens to indicate the start and end of the sequence. Subsequently, $\{f_t^{\text{sos}}, f_t^1 \dots f_t^{\text{eos}}\}$ are fed to transformer block [32], which employs masked self-attention to explore relationships between blocks. Finally, all the textual features $\{f_t^{\text{sos}}, f_t^1 \dots f_t^{\text{eos}}\}$ undergo linear projection, where $f_t^{\text{eos}}$ is transformed into the textual global feature $f_t^g$, and the others represent the textual local features:

$$f_t^g, f_t^1 \dots f_t^M = \phi(T), \quad (2)$$

where $\phi$ represents text encoder of CLIP.

*2.1.3 Keyword Statistics and Mask Generation.* We initially perform a statistical analysis to identify key concepts that require masking. Through word frequency analysis across the entire dataset, we exclude common high-frequency words such as "a", "the", "of", etc. Subsequently, we select top-$k$ frequency keywords in each dataset, yielding the corresponding keyword list. The process of keyword statistics can be summarized as follows:

$$List_{key} = Top_k\{Frequency(\sum_{i=1}^M T_i)\}, \quad (3)$$

We merge the keyword lists from each datset and remove any duplicate words across them, resulting in the final keyword list for training. If a word in the input text matches a word in the keyword list, it is replaced with "$[mask]$". Accordingly, we generate the masked text $T_{mask}$. Subsequently, we input the sentences after

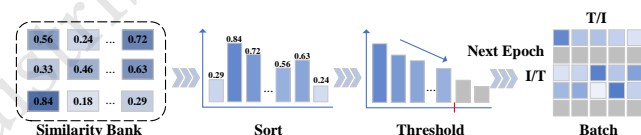

**Figure 4: Explanation of Eliminate Before Align strategy. 1) Initially, a similarity bank is established to store all global similarities on-the-fly. 2) Sort all the similarities. 3) Set the threshold based on the drop ratio. 4) Eliminate rows within a batch corresponding to Image-to-Text or Text-to-Image pairs with similarities below the threshold before the alignment in the next epoch.**

masking into the text encoder, obtaining the corresponding masked global feature $f_m^g$ and local features $\{f_m^1, f_m^2 \dots f_m^M\}$ :

$$f_m^g, f_m^1 \dots f_m^M = \phi(T_{mask}). \quad (4)$$

## 2.2 Eliminate Before Align

First, we conduct global alignment between the visual global feature $f_v^g$ and the textual global feature $f_t^g$. Similar to CLIP [29], we compute the global cosine similarity $Sim^g = cos\left(f_v^g, f_t^g\right)$.

Then, we conduct fine-grained alignment on the local features obtained from the visual and textual encoders. For each input image $I$ and text $T$, we obtain visual local features $\{f_v^1 \dots f_v^N\}$ and textual local features $\{f_t^1 \dots f_t^N\}$. We then compute the local cosine similarity for each local block $Sim_{ij} = cos\left(f_v^i, f_t^j\right)$. Next, we

obtain the corresponding local similarities for each image-text pairs by performing two consecutive L2-norm operations on the local blocks:

$$Sim^l = \left\| Sim_{ij} \right\|_{2,2} \tag{5}$$

where $\|\cdot\|_{2,2}$ represents the consecutive application of the L2-norm twice. During the inference stage, we combine the global similarity and local similarity via a weighted approach to obtain the final similarity between image and text:

$$Sim = \alpha Sim^g + \beta Sim^l. \tag{6}$$

Next, we introduce our proposed EBA strategy as shown in Figure 4. Specifically, we begin with conducting several epochs of regular training without eliminating any training samples, allowing the model to encounter all training samples during this process. Subsequently, we establish the Similarity Bank (*SimBank*), wherein we record the global similarity scores of all sample pairs within the current epoch:

$$Simbank = \left\{ Sim_i^g \right\}_{i=1}^L, \tag{7}$$

where $Sim_i^g$ represents the $i$-th global similarity, $L$ is the total number of image-text pairs in the dataset. Upon completion of an epoch, we extract all similarity values and sort them in descending order. We then select the similarity from the end of the sorted list by the predetermined drop ratio of the data volume, which will be served as the threshold $Th$ for the next epoch:

$$Th = Sort \left( Simbank \right) \left[ drop_{ratio} \right], \tag{8}$$

where $Sort$ indicates sorting in descending order and $drop_{ratio}$ represents the specified elimination rate. During the training of the next epoch, the similarity of all matched image-text pairs within each batch is compared with the threshold $Th$. If the similarity of the current image-text sample pair does not exceed the threshold $Th$, its loss is excluded from the current batch. Suppose there are $M$ instances within a batch that do not exceed the threshold. When calculating the loss for image-to-text and text-to-image, the corresponding rows are removed, transforming the $N \times N$ matrix into an $(N - M) \times N$ matrix before alignment:

$$B_s^g = \left\{ \sum_i^N Sim_i^g \middle| Sim_i^g > Th \right\} = \sum_i^{N-M} Sim_i^g, \tag{9}$$

where $Sim_g^i$ represents the global similarities in the $i$-th row, $B_g^s$ represents the global similarity matrix within a batch. If a global similarity is eliminated, the corresponding local similarity will also be eliminated. Similarly, the formulation can be expressed as:

$$B_s^l = \left\{ \sum_i^N Sim_i^l \middle| Sim_i^g > Th \right\} = \sum_i^{N-M} Sim_i^l. \tag{10}$$

## 2.3 Keyword Explicit Reasoning

Following [12], we adopt a single cross-attention layer, along with four Transformer blocks and a final Masked Language Modeling (MLM) head, to construct the word reasoning architecture. However, we observe that employing implicit reasoning on random tokens might overlook the subtle yet crucial differences among remote sensing images, which will be deeply analyzed in Section 3.5.3. Therefore, we adopt keyword explicit reasoning module, leveraging the key concepts extracted through Keyword Statistics and Mask

Generation, to explicitly incorporate meaningful keywords into the fine-grained contrastive learning process.

First, the visual features $\left\{ f_v^g, f_v^1 \dots f_v^N \right\}$ are served as $\mathcal{K}$ and $\mathcal{V}$, while masked textual features $\left\{ f_m^1 \dots f_m^M \right\}$ are served as $\mathcal{Q}$, yielding the corresponding predicted probability:

$$\left\{ p_i^m | m \in List_{key} \right\}_{i=1}^M = MLM_{head} \left( Transformer_N \left( CA \left( \mathcal{Q}, \mathcal{K}, \mathcal{V} \right) \right) \right), \tag{11}$$

where $CA$ represents cross attention layer, which reasons the relationship between $\mathcal{Q}, \mathcal{K},$ and $\mathcal{V}, Transformer_N$ represents $N$ Transformer blocks in accordance with CLIP [29], $MLM_{head}$ represents an MLP structure composed of a linear layer, QuickGELU layer, LayerNorm layer, and another linear layer. $\left\{ p_i^m | m \in List_{key} \right\}_{i=1}^M$ denotes the predicted probability $p$ at position $i$ for the mask $m$ of the $List_{key}$. The loss function is defined as follows:

$$\mathcal{L}_{mlm} = -\frac{1}{MV} \sum_{m=1}^M \sum_{i=1}^V y_i^m \log \frac{\exp \left( p_i^m \right)}{\sum\limits_{j=1}^V \exp \left( p_j^m \right)}, \tag{12}$$

where $M$ represents the number of masked tokens, $V$ is the vocabulary size of CLIP, $y_i^m$ is the one-hot distribution of the $m$-th masked word in the vocabulary of CLIP corresponding to the $i$-th token.

## 2.4 Loss Function

We employ the prevalant InfoNCE loss [26] to achieve fine-grained alignment, applied to both global and local similarities, which can be expressed as follows:

$$\mathcal{L}_{info} = -\frac{1}{N} \sum_{j=1}^N \left( log \frac{exp \left( s_j^{vt^+}/\gamma \right)}{\sum_{i=1}^N exp \left( s_{ij}^{vt}/\gamma \right)} - log \frac{exp \left( s_j^{tv^+}/\gamma \right)}{\sum_{i=1}^N exp \left( s_{ij}^{tv}/\gamma \right)} \right), \tag{13}$$

where $s^{vt^+}$ and $s^{tv^+}$ represent the positive pairs, $\sum_{i=1}^N s_{ij}^{vt}$ and $\sum_{i=1}^N s_{ij}^{tv}$ respectively represent the sum of each row in the similarity matrices for Image-to-Text or Text-to-Image alignments, $\gamma$ represents the temperature hyper-parameter, $N$ represents the batch size. For the matrices corrected by the EBA strategy during training, we eliminate the corresponding noisy image-text pairs and make the following adjustments to the InfoNCE loss:

$$\widetilde{\mathcal{L}}_{info} = -\frac{1}{N} \sum_{j=1}^N \left( log \frac{exp \left( s_j^{vt^+}/\gamma \right)}{\sum_{i=1}^{N-M} exp \left( s_{ij}^{vt}/\gamma \right)} - log \frac{exp \left( s_j^{tv^+}/\gamma \right)}{\sum_{i=1}^{N-M} exp \left( s_{ij}^{tv}/\gamma \right)} \right), \tag{14}$$

where $\sum_{i=1}^{N-M} s_{ij}^{vt}$ and $\sum_{i=1}^{N-M} s_{ij}^{tv}$ respectively represent the sum of each row in the similarity matrices for Image-to-Text or Text-to-Image after removing $M$ rows.

Both global and local alignment utilize the InfoNCE loss, while the modeling of masked attributes employs the MLM loss. We set a drop epoch $K$, before which we employ the original InfoNCE loss to expose the model to all data. Once the epoch exceeds $K$, we switch to the modified InfoNCE loss to eliminate noise. The overall loss function is formulated as:

$$\mathcal{L}_{total} = \begin{cases} \mathcal{L}_{info}^g + \mathcal{L}_{info}^l + \alpha \mathcal{L}_{mlm}, & if \ epoch < K \\ \widetilde{\mathcal{L}}_{info}^g + \widetilde{\mathcal{L}}_{info}^l + \alpha \mathcal{L}_{mlm}, & if \ epoch \geq K \end{cases} \tag{15}$$

**Table 1: Comparison results of the image-text retrieval on RSICD, RSITMD, and NWPU.**

| Approach | Backbone | Train Set | Test Set | Image-to-Text Retrieval | | | Text-to-Image Retrieval | | | mR |
|---|---|---|---|---|---|---|---|---|---|---|
| | | | | R@1 | R@5 | R@10 | R@1 | R@5 | R@10 | |
| VSE++$_{BMVC2018}$ [9] | ResNet18/Bi-GRU | RSICD | RSICD | 3.38 | 9.51 | 17.46 | 2.82 | 11.32 | 18.10 | 10.43 |
| LW-MCR$_{TGRS2021}$ [39] | ResNet18/Bi-GRU | RSICD | RSICD | 3.29 | 12.52 | 19.93 | 4.66 | 17.51 | 30.02 | 14.66 |
| AMFMN$_{TGRS2022}$ [38] | ResNet18/Bi-GRU | RSICD | RSICD | 5.39 | 15.08 | 23.40 | 4.90 | 18.28 | 31.44 | 16.42 |
| GaLR$_{TGRS2022}$ [41] | ResNet18/Bi-GRU | RSICD | RSICD | 6.59 | 19.85 | 31.04 | 4.69 | 19.48 | 32.13 | 18.96 |
| KCR$_{IJCAI2022}$ [25] | ResNet101/Bert | RSICD | RSICD | 5.95 | 19.59 | 29.58 | 5.40 | 22.44 | 37.36 | 19.89 |
| Multilanguage$_{JSTARS2022}$ [2] | ViT-B-32/Bert | RSICD | RSICD | 10.70 | 29.64 | 41.53 | 9.14 | 28.96 | 44.59 | 27.42 |
| SWAN$_{ICMR2023}$ [28] | ResNet50/Glove | RSICD | RSICD | 7.41 | 20.13 | 30.86 | 5.56 | 22.26 | 37.41 | 20.61 |
| FAMMI$_{TGRS2023}$ [46] | DetNet/Bert | RSICD | RSICD | 10.44 | 22.66 | 30.89 | 8.11 | 25.59 | 41.37 | 23.18 |
| PIR$_{ACMMM2023}$ [27] | Swin-T/Bert | RSICD | RSICD | 9.88 | 27.26 | 39.16 | 6.97 | 24.56 | 38.92 | 24.46 |
| KAMCL$_{TGRS2023}$ [11] | ResNet101/Bi-GRU | RSICD | RSICD | 12.08 | 27.26 | 38.70 | 8.65 | 27.43 | 42.51 | 26.10 |
| PE-RSITR$_{TGRS2023}$ [37] | CLIP(ViT-B-32) | RSICD | RSICD | 14.13 | 31.51 | 44.78 | 11.63 | 33.92 | 50.73 | 31.12 |
| RemoteCLIP$_{arxiv2023}$ [21] | CLIP(ViT-B-32) | RET-3+DET-10+SEG-4 (0.82M) | RSICD | 17.02 | 37.97 | 51.51 | 13.71 | 37.11 | 54.25 | 35.26 |
| GeoRSCLIP$_{arxiv2023}$ [45] | CLIP(ViT-B-32) | RS5M+RET-2(5M+0.07M) | RSICD | 21.13 | 41.72 | 55.63 | 15.59 | 41.19 | 57.99 | 38.87 |
| **EBAKER(Ours)** | CLIP(ViT-B-32) | RSICD+RSITMD+NWPU(0.2M) | RSICD | **21.87** | **44.46** | **58.92** | **17.37** | **43.00** | **58.55** | **40.70** |
| VSE++$_{BMVC2018}$ [9] | ResNet18/Bi-GRU | RSITMD | RSITMD | 10.38 | 27.65 | 39.60 | 7.79 | 24.87 | 38.67 | 24.83 |
| LW-MCR$_{TGRS2021}$ [39] | ResNet18/Bi-GRU | RSITMD | RSITMD | 10.18 | 28.98 | 39.82 | 7.79 | 30.18 | 49.78 | 27.79 |
| AMFMN$_{TGRS2022}$ [38] | ResNet18/Bi-GRU | RSITMD | RSITMD | 11.06 | 29.20 | 38.72 | 9.96 | 34.03 | 52.96 | 29.32 |
| GaLR$_{TGRS2022}$ [41] | ResNet18/Bi-GRU | RSITMD | RSITMD | 14.82 | 31.64 | 42.48 | 11.15 | 36.68 | 51.68 | 31.41 |
| Multilanguage$_{JSTARS2022}$ [2] | ViT-B-32/Bert | RSITMD | RSITMD | 19.69 | 40.26 | 54.42 | 17.61 | 49.73 | 66.59 | 41.38 |
| SWAN$_{ICMR2023}$ [28] | ResNet50/Glove | RSITMD | RSITMD | 13.35 | 32.15 | 46.90 | 11.24 | 40.40 | 60.60 | 34.11 |
| FAMMI$_{TGRS2023}$ [46] | DetNet/Bert | RSITMD | RSITMD | 16.15 | 35.62 | 48.89 | 12.96 | 42.39 | 59.96 | 35.99 |
| PIR$_{ACMMM2023}$ [27] | Swin-T/Bert | RSITMD | RSITMD | 18.14 | 41.15 | 52.88 | 12.17 | 41.68 | 63.41 | 38.24 |
| KAMCL$_{TGRS2023}$ [11] | ResNet101/Bi-GRU | RSITMD | RSITMD | 16.51 | 36.28 | 49.12 | 13.50 | 42.15 | 59.32 | 36.14 |
| PE-RSITR$_{TGRS2023}$ [37] | CLIP(ViT-B-32) | RSITMD | RSITMD | 23.67 | 44.07 | 60.36 | 20.10 | 50.63 | 67.97 | 44.47 |
| RemoteCLIP$_{arxiv2023}$ [21] | CLIP(ViT-B-32) | RET-3+DET-10+SEG-4(0.82M) | RSITMD | 27.88 | 50.66 | 65.71 | 22.17 | 56.46 | 73.41 | 49.38 |
| GeoRSCLIP$_{arxiv2023}$ [45] | CLIP(ViT-B-32) | RS5M+RET-2(5M+0.07M) | RSITMD | 32.30 | 53.32 | 67.92 | 25.04 | 57.88 | 74.38 | 51.81 |
| **EBAKER(Ours)** | CLIP(ViT-B-32) | RSICD+RSITMD+NWPU(0.2M) | RSITMD | **34.07** | **54.20** | **67.95** | **28.05** | **60.35** | **75.31** | **53.32** |
| VSE++$_{BMVC2018}$ [9] | ResNet18/Bi-GRU | NWPU | NWPU | 4.84 | 12.89 | 20.94 | 4.38 | 13.61 | 24.12 | 13.46 |
| AMFMN$_{TGRS2022}$ [38] | ResNet18/Bi-GRU | NWPU | NWPU | 11.49 | 38.75 | 57.73 | 8.63 | 30.25 | 46.48 | 32.22 |
| KAMCL$_{TGRS2023}$ [11] | ResNet101/Bi-GRU | NWPU | NWPU | 21.02 | 57.33 | 74.41 | 12.74 | 38.03 | 53.90 | 42.90 |
| RemoteCLIP$_{arxiv2023}$ [21] | CLIP(ViT-B-32) | RET-3+DET-10+SEG-4+NWPU(0.97M) | NWPU | 24.57 | 57.75 | 74.19 | **14.95** | 40.17 | 55.75 | 44.56 |
| **EBAKER(Ours)** | CLIP(ViT-B-32) | RSICD+RSITMD+NWPU(0.2M) | NWPU | **24.98** | **60.95** | **77.68** | 14.55 | **41.16** | **56.77** | **46.02** |

## 3 EXPERIMENTS

### 3.1 Datasets

In our experiments, we employ three benchmark datasets, RSICD [23], RSITMD [38], and NWPU [5], to validate the effectiveness of our approach. Following the methodology of RemoteCLIP [21], we generate $p$-Hash values for image-text pairs and set a threshold of 2 to merge these three datasets into one, aiming to eliminate redundant images. The RSICD dataset comprises 10,921 images, each sized at 224×224 pixels, making it the most widely utilized dataset for RSITR. RSITMD consists of 4,743 images, each with a size of 256×256 pixels. The NWPU dataset encompasses 31,500 images, also with a size of 256×256 pixels. Following the protocol of [38], we divide these three datasets into train sets (80%), validation sets (10%), and test sets (10%).

### 3.2 Metrics

We utilize Recall at $k$ ($R@k$, $k$=1,5,10) and mean Recall (mR) as the evaluation metrics to evaluate the retrieval performance. Particularly, $R@k$ measures the percentage of ground truth instances within the top $k$ samples, while mR represents the average value across all six $R@k$ metrics, providing an overall assessment of retrieval performance.

### 3.3 Implementation Details

The implementation of EBAKER is derived from the RemoteCLIP codebase [21], with the ViT-B-32 architexture provided by Open-CLIP [6]. We train the model for 7 epochs with a batch size of 100 and employ Adam [13] as our optimizer. Additionally, we adopt linear warm-up and cosine learning rate scheduler. The learning rate is set to 1.5e-5 and weight decay is set to 0.7, warmup is set to 200, maximum gradient norm is set to 50. For EBA strategy, we set drop epoch $K$ to 4 and drop ratio to 1%. KER transformer block $N$ is set to 4. All the experiments are implemented with PyTorch and trained with a single NVIDIA GeForce RTX 4090 GPU.

### 3.4 Comparisons with the SOTA Methods

We compare our proposed EBAKER with existing state-of-the-art methods, which include traditional methods like VSE++ [9], LW-MCR [39], AMFMN [38], GaLR [41], KCR [25], Multilanguage [2], SWAN [28], FAMMI [46], PIR [27], and KAMCL [11]. Additionally, we cover CLIP-based methods such as PE-RSITR [37], RemoteCLIP [21], and GeoRSCLIP [45]. Our proposed EBAKER falls under the category of CLIP-based methods. The experimental results for the RSICD [23], RSITMD [38], and NWPU [5] datasets are presented in Table 1. Additionally, the vision and text backbones (separated by

**Table 2: Ablation experiments with different modules of on RSICD test set**

| Module | Image-to-Text Retrieval | | | Image-to-Text Retrieval | | | mR |
|---|---|---|---|---|---|---|---|
| | R@1 | R@5 | R@10 | R@1 | R@5 | R@10 | |
| baseline | 18.85 | 39.35 | 54.08 | 15.93 | 41.45 | 57.85 | 37.92 |
| +local | 20.95 | 42.45 | 56.45 | 16.51 | 41.99 | 57.77 | 39.35 |
| +KER | 19.58 | 42.63 | 56.72 | 16.61 | 42.20 | 58.23 | 39.33 |
| +EBA | 19.58 | 41.72 | 56.36 | 16.98 | 43.18 | 58.19 | 39.34 |
| +local+KER | 21.23 | 43.92 | 58.37 | 17.18 | 43.22 | 58.79 | 40.45 |
| +local+EBA | 20.59 | **44.65** | 57.64 | 17.24 | 42.29 | 57.93 | 40.05 |
| +KER+EBA | 20.85 | 42.99 | 56.81 | **17.71** | **43.27** | 57.94 | 39.93 |
| +local+KER+EBA | **21.87** | 44.46 | **58.92** | 17.37 | 43.00 | **58.55** | **40.70** |

"/"), total train set, and test set are provided within the table. From these results, we draw the following observations and conclusions.

*3.4.1 Quantitative Comparison on RSICD, RSITMD, and NWPU.* For the RSICD dataset, our EBAKER method notably outperforms all competing methods across various evaluation metrics. Specifically, compared with GeoRSCLIP [45], our EBAKER method surpasses it on all evaluation metrics, while utilizing only 0.2 million data samples, a mere 4% of the 5.07 million utilized by GeoRSCLIP. Particularly noteworthy is the 3.29% improvement in image-to-text R@10, a 1.78% enhancement in text-to-image R@1, and an overall mR increase of 1.83%. For the RSITMD dataset, also compared with GeoRSCLIP, we achieve a 1.77% improvement in image-to-text R@1, a 3.01% enhancement in text-to-image R@1, and an overall mR increase of 1.51%. These results demonstrate a comprehensive performance superiority over GeoRSCLIP. Following [11], we also conduct comparative experiments on the NWPU dataset, where the results of RemoteCLIP [21] are reproduced by fine-tuning on NWPU by the code and weight files provided in [21]. The results indicate that our method achieves superior performance with less data. Specifically, we observe a 3.49% improvement in image-to-text R@10, a 1.02% enhancement in text-to-image R@10, and an overall mR increase of 1.46%.

*3.4.2 Comparison between Traditional and CLIP-based Method.* As indicated by the backbones specified in Table 1, we categorize the methods into traditional approaches and CLIP-based methods. Compared with traditional approaches, CLIP-based methods often achieve better performance through fine-tuning. However, they typically require more training data. For instance, GeoRSCLIP[45] necessitates an additional 5M remote sensing dataset for the process of RS pretraining, as shown in Figure 2 (a). Our proposed EBAKER method achieves a balance between performance and computational cost by solely relying on the combination of the RSICD, RSITMD, and NWPU datasets, without the need for remote sensing data for an extra RS pretraining process. Compared with the traditional method KAMCL [11], we achieve performance improvements of 14.60% and 17.18% in mR on the RSICD and RSITMD datasets, respectively. In comparison to the CLIP-based method GeoRSCLIP[45], we only need a one-step fine-grained training, thereby eliminating the need for additional pretraining samples, and achieving performance improvements of 1.83% and 1.51% in mR, respectively.

**Table 3: Ablation on the ratio of global and local alignment on RSICD test set**

| Method. | Global | Local | Image-to-Text Retrieval | | | Image-to-Text Retrieval | | | mR |
|---|---|---|---|---|---|---|---|---|---|
| | | | R@1 | R@5 | R@10 | R@1 | R@5 | R@10 | |
| 1 | 1 | 0 | 22.32 | 44.10 | 58.01 | 16.07 | 42.07 | 58.06 | 40.10 |
| 2 | 0.9 | 0.1 | **22.96** | 43.82 | 58.37 | 16.45 | 42.29 | 58.50 | 40.40 |
| 3 | 0.8 | 0.2 | 22.78 | 44.46 | 58.37 | 16.93 | 42.36 | 58.48 | 40.56 |
| 4 | 0.7 | 0.3 | 22.14 | 44.28 | 58.28 | 17.20 | 42.84 | **59.65** | 40.56 |
| 5 | 0.6 | 0.4 | 21.87 | 44.46 | **58.92** | 17.37 | **43.00** | 58.55 | **40.70** |
| 6 | 0.5 | 0.5 | 21.87 | **44.74** | 58.65 | 17.35 | 42.80 | 58.41 | 40.63 |
| 7 | 0.4 | 0.6 | 21.32 | 44.65 | 58.83 | **17.47** | 42.73 | 58.46 | 40.58 |
| 8 | 0.3 | 0.7 | 21.04 | 44.28 | 58.74 | 17.42 | 42.63 | 58.39 | 40.42 |
| 9 | 0.2 | 0.8 | 21.04 | 43.92 | 58.37 | 17.24 | 42.58 | 58.33 | 40.25 |
| 10 | 0.1 | 0.9 | 21.13 | 43.73 | 58.28 | 17.13 | 42.62 | 58.12 | 40.17 |
| 11 | 0 | 1 | 21.23 | 43.82 | 58.28 | 17.13 | 42.62 | 58.04 | 40.19 |

**Table 4: Different mask strategies on RSICD test set**

| Approach | Image-to-Text Retrieval | | | Image-to-Text Retrieval | | | mR |
|---|---|---|---|---|---|---|---|
| | R@1 | R@5 | R@10 | R@1 | R@5 | R@10 | |
| IRR[12] | 20.04 | 41.81 | 55.35 | 16.93 | 42.93 | 58.30 | 39.23 |
| MAM | 20.68 | 43.00 | 56.91 | 16.85 | 42.03 | **58.68** | 39.69 |
| KER | **21.87** | **44.46** | **58.92** | **17.37** | **43.00** | 58.55 | **40.70** |

## 3.5 Ablation Studies

In this section, we design a variety of ablation experiments, which aim at investigating the performance gains of different modules in the model to verify the effectiveness of each part.

*3.5.1 Ablation Studies of Structures.* As shown in Table 2, we conduct ablation studies on different modules to demonstrate the effectiveness of our method. Initially, we choose the original CLIP as the baseline, and incorporate local alignment (local), the KER module, and the EBA strategy, respectively. Compared with the baseline, these additions resulted in respective improvements of 1.43%, 1.41%, and 1.42% in terms of mR. Subsequently, we conduct the combinations of each two modules and find that local alignment with KER module yields promising results, with an improvement of approximately 1.10% compared with utilizing either mechanism individually. This may be attributed to the fact that the reasoning ability of key concepts of KER explicitly manifests in the fine-grained local alignment. Ultimately, the integration of all modules achieved an mR of 40.70%.

*3.5.2 Ablation on the Ratio of Global and Local Alignment.* To further explore the impact of global and local alignment, we conduct additional ablation experiments by varying the weights of global and local alignment, as shown in Table 3. We maintain the sum of the weights for global and local alignment to be 1. According to the results, the combination of global and local alignment performs best when the weights are set to 0.6 and 0.4, respectively. In this configuration, the mR reach 40.70%, which is 0.60% higher than utilizing only global features and 0.51% higher than utilizing only local features. This demonstrates that global and local information complement each other, allowing for fine-grained contrastive learning to better capture and distinguish the details within remote sensing images.

**Table 5: Ablation on hyperparameter on objective function on RSICD test set**

| $\alpha$(MLM) | Image-to-Text Retrieval | | | Image-to-Text Retrieval | | | mR |
|---|---|---|---|---|---|---|---|
| | R@1 | R@5 | R@10 | R@1 | R@5 | R@10 | |
| 0.1 | 21.23 | 43.55 | 55.17 | 16.51 | **43.28** | 58.16 | 39.65 |
| 0.2 | 19.76 | 42.18 | 56.08 | **17.77** | 43.07 | 58.72 | 39.60 |
| 0.3 | 20.59 | 43.55 | 58.01 | 17.42 | 42.96 | **59.01** | 40.26 |
| 0.4 | 21.77 | 43.28 | 58.54 | 17.26 | 42.87 | 58.23 | 40.33 |
| 0.5 | **21.87** | **44.46** | **58.92** | 17.37 | 43.00 | 58.55 | **40.70** |
| 0.6 | **21.87** | 43.37 | 57.37 | 17.09 | 43.09 | 58.72 | 40.25 |
| 0.7 | 21.42 | 43.54 | 56.81 | 17.11 | 42.58 | 58.76 | 40.04 |
| 0.8 | 21.59 | 43.73 | 57.18 | 16.71 | 43.04 | 58.81 | 40.18 |
| 0.9 | 21.13 | 40.71 | 56.27 | 17.53 | 43.06 | 58.99 | 39.62 |
| 1 | 19.12 | 40.26 | 55.63 | 16.72 | 42.40 | 58.59 | 38.79 |

*3.5.3 Ablation on Different Mask Strategies.* To validate the effectiveness of our proposed Keyword Explicit Reasoning (KER) module, we conduct comparisons with similar algorithms as shown in Table 4. Implicit Relation Reasoning (IRR) [12] utilizes a masked language modeling (MLM) approach similar to BERT [8] to implicitly aggregate vision and text features, yielding promising results. However, we argue that this random masking approach may not effectively capture key concepts, as common words like "is" and "a" are also masked. Predicting such words do not significantly contribute to the ability of truly subtle distinctions in text. Thus, we first devise a Masked Attribute Modeling (MAM) module specifically tailored for attribute words, as these terms often offer more discriminative information in retrieval tasks. We annotate and mask all adjectives in the corpus to predict them. In the experiment, we find that this MAM module results in a 0.46% improvement in mR compared with IRR. Subsequently, we further propose our KER module targeted at key concepts. Finally, compared with IRR, our results demonstrate a 1.47% increase in mR, highlighting the importance of key concepts in RSITR tasks and the effectiveness of KER module.

*3.5.4 Ablation on Hyperparameter on Objective Function.* As shown in Table 5, we further investigate the hyperparameters of the loss function. While keeping the ratio of global and local loss functions constant, we adjust the weight of the MLM component. The results indicate that the optimal weight for the MLM loss component is 0.5. It is important to note that the weight assigned to MLM should not be excessively high, as the primary focus of the model remains on the process of contrastive learning, and predicting keywords serves as a secondary objective.

*3.5.5 Ablation on Drop Epoch and Drop Ratio.* We also conduct an ablation study on the drop epoch as shown in Figure 5 (a). Given that our training comprises a total of 7 epochs, we vary the drop epoch $K$ for the EBA strategy from 1 to 7. Before the drop epoch, the model encounters the entire dataset. The ultimate results of the ablation study suggest that the most effective drop epoch is during the 4th epoch, with an increase of 0.79% in mR.

Moreover, the ablation studies on the drop ratio are shown in Figure 5 (b) to investigate its impact on the EBA strategy. The results indicate that when the drop ratio is set to 1%, which means that setting the lowest 1% similarity values stored in the sim bank as the

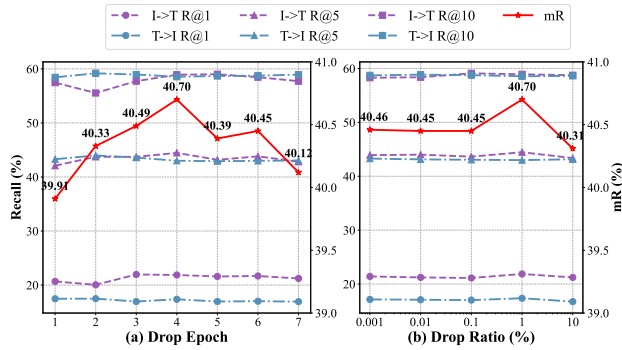

**Figure 5: Ablation on drop epoch and drop ratio. R@k (k=1,5,10) corresponds to the left vertical axis, while meanR corresponds to the right vertical axis.**

**Table 6: Ablation on the quantity of keywords for each dataset on RSICD test set**

| Total | Word | Image-to-Text Retrieval | | | Image-to-Text Retrieval | | | mR |
|---|---|---|---|---|---|---|---|---|
| | | R@1 | R@5 | R@10 | R@1 | R@5 | R@10 | |
| 27 | 16 | 19.67 | 40.90 | 55.17 | 17.05 | 42.27 | 58.24 | 38.88 |
| 57 | 32 | 20.59 | 41.99 | 57.18 | 16.98 | 43.81 | **59.16** | 39.95 |
| 109 | 64 | 20.86 | 43.00 | 56.63 | 17.69 | 42.34 | 57.90 | 39.74 |
| 198 | 128 | 20.59 | 42.18 | 57.09 | 16.85 | 42.73 | 58.68 | 39.69 |
| 394 | 256 | **22.32** | 43.64 | 56.63 | **17.82** | **43.77** | 58.98 | 40.53 |
| 800 | 512 | 21.87 | **44.46** | **58.92** | 17.37 | 43.00 | 58.55 | **40.70** |

threshold for filtering the global similarity for the next epoch, it effectively filters out noisy sample pairs. However, when the drop ratio is low, the EBA strategy filters out too few samples, failing to effectively eliminate noise. Conversely, when the drop ratio is high, a large number of normal samples are filtered out, leading to suboptimal performance. Therefore, based on these findings, we determine the drop ratio to be 1%, striking a balance between noise elimination and retaining sufficient samples for effective training.

*3.5.6 Ablation on the Quantity of Keywords for Each Dataset.* We further conduct an ablation study on the number of keywords obtained through word frequency analysis on the dataset, as shown in Table 6. Here, "Word" represents the number of words selected based on the highest frequency for each dataset. "Total" indicates the total number of unique keywords obtained by merging and removing duplicates across all three datasets. The results suggest that the number of words selected for the keyword list should be within an appropriate range. Ultimately, we select the top 512 words with the highest frequency for each dataset as the keyword list.

## 3.6 Trade-off between Model Performance and Inference Speed.

In our comparison of inference time across various approaches, we conduct traditional and CLIP-based methods across the three datasets with a single NVIDIA GeForce RTX 4090 GPU. The "IT" in table represents Inference Time. Note that the GaLR [41] method are not replicated on the NWPU dataset due to a lack of detailed

## Image-to-Text Retrieval

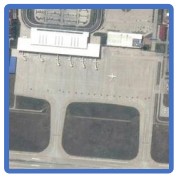

1.There is a row of red roofed houses near the airport .
2.There is a row of red roofed houses near the airport .
3.The airport has a lot of white planes .
4.There are a lot of bare land around the airport .
5.A tall airport was built on the land .

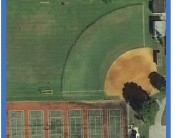

1.A row of tennis courts are near a baseball field .
2.There is a gray room next to a baseball field.
3.There is a gray room next to the baseball field.
4.There are six tennis courts next to the baseball field.
5.four baseball fields are surrounded by trees and meadows .

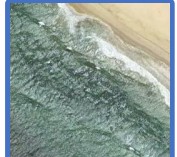

1.Green clear sea and light yellow beach .
2.The beach with khaki sand and the color of the seawater is green .
3.The beach with khaki sand and the color of the seawater is green .
4.Green water and sandy beaches .
5.The waves of the green seawater wash the beach .

## Text-to-Image Retrieval

A viaduct in the shape of figure of eight is on the bareland .

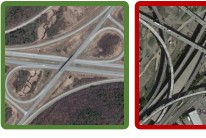 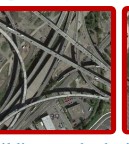 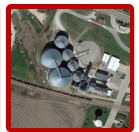 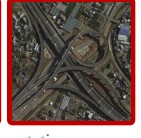 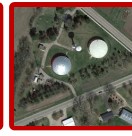

There are many buildings on both sides of the railway station.
There is a long blue building.

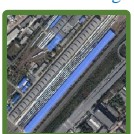 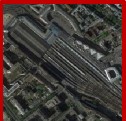 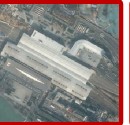 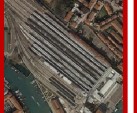 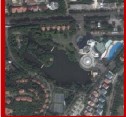

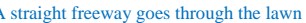

A straight freeway goes through the lawn .

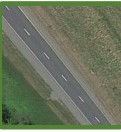 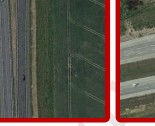 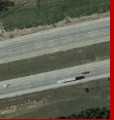 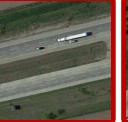 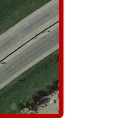

**Figure 6: Visualization of the top 5 image-to-text and text-to-image retrieval results. Each row corresponds to the outcomes obtained from RSICD [23], RSITMD [38], and NWPU [5] datasets, respectively. Queries are denoted in blue, correct results in green, and wrong results in red.**

**Table 7: Trade-off between model performance and inference speed.**

| Approach | RSICD | | RSITMD | | NWPU | |
|---|---|---|---|---|---|---|
| | mR | IT(s) | mR | IT(s) | mR | IT(s) |
| VSE++[9] | 10.43 | 8.63 | 24.83 | 5.52 | 13.46 | 22.68 |
| AMFMN[38] | 16.42 | 25.56 | 29.72 | 6.39 | 32.22 | 148.41 |
| GaLR[41] | 18.96 | 22.92 | 31.41 | 6.23 | - | - |
| KAMCL[11] | 23.26 | 11.86 | 36.19 | 5.63 | 40.75 | 28.53 |
| RemoteCLIP[21] | 35.26 | 2.42 | 49.38 | 1.42 | 42.90 | 6.38 |
| **EBAKER(Ours)** | **40.70** | 5.96 | **53.32** | 2.53 | **46.02** | 20.30 |

information regarding the additional Ppyolo extractor [22] it employs. As shown in Table 7, the results indicate that our model lags behind that of the RemoteCLIP, which relies on excluding global features, in terms of inference speed. This discrepancy arises from the integration of fine-grained local alignment in our model, leading to a requirement for increased inference time. Despite this, our proposed EBAKER outperforms RemoteCLIP by 5.30%, 3.84%, and 3.12% in terms of performance, albeit with an increase in inference time by 146%, 78%, and 218%, respectively. Nevertheless, when compared with traditional methods such as KAMCL [11] and GaLR [41], we exhibit significant advantages in both performance and inference time.

### 3.7 Visualization of Results

Figure 6 displays the top 5 retrieval results, with each row showing the results on the RSICD [23], RSITMD [38], and NWPU [5] datasets, respectively. As depicted, our proposed EBAKER successfully retrieves the target images under given queries. Additionally, examining the incorrect retrieved results in the first and third rows

reveal that they are actually highly relevant to the query image. This suggests that weakly correlated image-text pairs not only exist in the train set but also in the validation and test sets. Therefore, such errors do not necessarily indicate that the model has failed to learn effective representations. In the future, we will investigate how to address such issue in the test set.

## 4 CONCLUSION

In this paper, we have introduced a novel Eliminate Before Align strategy with Keyword Explicit Reasoning framework (EBAKER), aiming at achieving the transfer of foundation model to remote sensing image-text retrieval model through a one-step fine-grained training. This framework incorporates an Eliminate Before Align strategy to eliminate weakly correlated pairs, thereby promoting the accuracy of fine-grained contrastive learning. Moreover, we employ a Keyword Explicit Reasoning module, which boosts the discriminative ability to the model by predicting nuanced differences in key concepts. Finally, we conduct comprehensive experiments to validate the superiority and effectiveness of our approach on three public benchmark datasets: RSICD, RSITMD, and NWPU.

Our method represents a novel attempt to skip the stage of RS pretraining, providing a promising solution for transferring large models directly to other tasks in the remote sensing domain. By tailoring the approach to the characteristics of different downstream tasks, we potentially save a significant amount of additional training data in various domains. Thus, it also lays a foundation for extending our method to other domains, such as product search and pedestrian retrieval for future development. In the future, our efforts will persist in researches on enabling models to adaptively filter data and pay attention to more fine-grained information, aiming at exploring the optimal application of multimodal foundation models in downstream tasks.

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
