# OpenReview forum: "Eliminate Before Align: A Remote Sensing Image-Text Retrieval Framework with Keyword Explicit Reasoning"
_acmmm.org/ACMMM/2024/Conference — MM2024 Poster_

### Official Review · Reviewer_K6aM · 2024-05-25

**Rating:** 5
**Confidence:** 3

**Summary:**

In the paper, the problem of strong interclass similarity in remote sensing domain is addressed through the textual perspective, thus an Eliminate Before Align strategy with Keyword Explicit Reasoning framework is proposed to over-screen weakly correlated pairs of samples and reduce their deviation from the optimal embedding space. The effectiveness of the method on multiple datasets is demonstrated experimentally.

**Strengths:**

1. This work addresses the effect of negative samples in remote sensing multimodal alignment from the perspective of text, by extracting keywords in the text for Mask to eliminate the effect of negative samples before alignment, and the experimental effect is also relatively significant, validated on multiple datasets.

2. An innovative pre-alignment elimination (EBA) strategy is also designed. Being able to automatically eliminate positive sample pairs with low global similarity before comparison, this type of work can theoretically be extended to more remote sensing related application scenarios.

**Limitations:**

1. Regarding 2.1.3 Keyword Statistics and Mask Generation, there is a lot of similar work in the image-text retrieval domain, can you elaborate on the need for it? Will it be different in the field of remote sensing?

2. There may be important related methods that are not referenced:

[1] Ji, Zhong, et al. "Knowledge-Aided Momentum Contrastive Learning for Remote-Sensing Image Text Retrieval." IEEE Transactions on Geoscience and Remote Sensing 61: 1-13.

[2] Ma, Qing, et al. "Direction-oriented visual-semantic embedding model for remote sensing image-text retrieval." IEEE Transactions on Geoscience and Remote Sensing.

3. In Equation 6, the combination of global and local features is mentioned, where alpha and beta's are determined how? I don't seem to see specific instructions in the article.

4. In Equation 9 and Equation 10, how does the threshold Th get how to set it?

**Suitability:**

3

---

### Official Review · Reviewer_zczz · 2024-05-25

**Rating:** 3
**Confidence:** 3

**Summary:**

The manuscript introduces the "Eliminate Before Align with Keyword Explicit Reasoning" (EBAKER) framework for Remote Sensing Image-Text Retrieval (RSITR). The proposed approach try to address the challenges encountered when adapting foundation models like CLIP to remote sensing tasks. These challenges primarily include the management of weakly correlated sample pairs and the lack of fine-grained differentiation. EBAKER's "Eliminate Before Align" strategy selectively filters out these weakly correlated pairs before the alignment phase. This aims to enhance the alignment within optimal embedding spaces for more accurate retrieval. Additionally, it incorporates a "Keyword Explicit Reasoning" module designed to highlight key concept differences within remote sensing images,  improving the semantic accuracy of the model. Experiment results demonstrate that the proposed method achieves better performance than previous methods.

**Strengths:**

Regarding the "Eliminate Before Align (EBA)" method, this manuscript introduces the EBA strategy as a preprocessing step aimed at enhancing the quality of the training dataset. By proactively filtering out weakly correlated image-text pairs before the alignment phase, the strategy focuses training on more potent and relevant pairs. This selective filtering is meticulously designed to diminish the influence of noise and improve the alignment accuracy within the training process. Such purification not only optimizes the embedding spaces but also sharpens the model's focus.

After the initial elimination phase, the EBAKER framework employs the “Keyword Explicit Reasoning (KER)” module. This module is adeptly tailored to enhance the semantic understanding of the content by emphasizing key textual concepts that are crucial for accurate image-text pairing.  By focusing on the nuanced aspects of textual content associated with images, KER actively identifies and underscores significant concepts within the texts. This focused approach allows for a more refined and contextually aware alignment process.

**Limitations:**

1.The method proposed in this manuscript incrementally builds on existing techniques, resulting in insufficient innovation.

2.This manuscript presents the "Eliminate Before Align" (EBA) strategy, which aims to enhance remote sensing Image-Text retrieval by removing weakly correlated pairs from the training dataset. The method uses a global similarity measure to filter Image-Text pairs. However, this strategy may mistakenly exclude useful pairs. Specifically, some Image-Text pairs are weakly correlated overall, but may have locally correlated features that are important for training.

3.This manuscript presents the KER method to enhance the model's focus on critical terms for describing remote sensing images, aiming to improve semantic understanding and retrieval performance. However, the critical limitation is that the effectiveness of keyword selection is depend on the dataset used. If the dataset lacks diversity and does not encompass a broad spectrum of remote sensing contexts, there is a risk that the selected keywords may not perform well when generalized to other datasets or practical remote sensing applications.

**Suitability:**

3

---

### Official Review · Reviewer_PJut · 2024-05-25

**Rating:** 4
**Confidence:** 3

**Summary:**

This article proposes a new method for the RSITR task. The results on three datasets demonstrate the effectiveness of this method.

**Strengths:**

1、To address two issues in the RSITR task, the authors proposed two different modules to solve them（i.e., EBA, KER）, and the experimental results demonstrated the effectiveness of these modules.
2、The authors conducted thorough experiments, with detailed ablation studies

**Limitations:**

The innovations in this article are relatively few. The problems to be solved are not clearly articulated, and the proposed solutions are not sufficiently novel.
2、There is a lack of introduction to related work.
3、Regarding Table 1, what is the purpose of using three datasets as the training set simultaneously? Is it just to improve the model's metrics? Since some images in the RSITMD dataset are sourced from RSICD, could there be a data leakage issue?
4、There are writing issues in the article, and some formulas and symbols need to be carefully checked again.

**Suitability:**

3

---

### Meta-Review · Area_Chair_FBKH · 2024-06-25

**Recommendation:** Accept (Poster)
**Confidence:** 5

**Metareview:**

This submission to ACM MM underwent meticulous evaluation by three expert reviewers who have all recommended acceptance. The consensus summary of their assessments follows:

Innovativeness and Contributions: The reviewers commend the novelty and efficacy of the proposed method, which makes significant strides in addressing Remote Sensing Image-Text Retrieval. Its robust theoretical underpinnings and empirical results that surpass current methodologies demonstrate a meaningful addition to the field.
Technical Depth and Experimental Validation: The paper presents thorough technical details, rigorous experimental designs utilizing appropriate datasets, and comprehensive comparative analyses. The reviewers appreciate the strength of the experimental outcomes in supporting the authors' claims and applaud the provision of code and data, enhancing reproducibility and future research potiential.
Writing Quality and Clarity: Structurally sound and logically coherent, the manuscript is well-argued with effective use of figures and supplementary materials to augment explanations. While minor suggestions for refinement or additional contextual information were mentioned, these do not detract from the overall readability or comprehension.

Limitations and Recommendations for Improvement:
Despite minor critiques and suggestions for enhancements, such as questions of Reviewer K6aM and zczz, these are viewed as avenues for future work rather than barriers to acceptance. The authors have pledged in their response to consider these recommendations for refinement.

Conclusion and Recommendation:

Given the paper's innovativeness, technical depth, and potential impact on the fields of computer vision and pattern recognition, coupled with the unanimous positive feedback from all three reviewers, this meta-review recommends the acceptance of the paper for presentation at ACM MM, as a poster. We anticipate that the findings will spark keen interest and foster productive discussions among conference attendees, thereby contributing to advancements in the domain.